# Persistent thermal input controls steering behavior in *Caenorhabditis elegans*

**Muneki Ikeda** [1,2,3]*, **Hirotaka Matsumoto** [4,5], **Eduardo J. Izquierdo** [6]

**1** Division of Biological Science, Graduate School of Science, Nagoya University, Nagoya, Aichi, Japan,
**2** Department of General Systems Studies, Graduate School of Arts and Sciences, The University of Tokyo,
Japan, **3** Department of Neurology, University of California San Francisco, San Francisco, California, United
States of America, **4** Laboratory for Bioinformatics Research RIKEN Center for Biosystems Dynamics
Research, Wako, Saitama, Japan, **5** School of Information and Data Sciences, Nagasaki University,
Nagasaki, Japan, **6** Cognitive Science Program, Indiana University, Bloomington, Indiana, United States of
America

* muneki.ikeda@ucsf.edu

journal.pcbi.1007916

Telecommunications Research Institute
International, JAPAN

**Data Availability Statement:** All relevant data are
within the manuscript, Supporting Information, and
our previous publication https://www.pnas.org/
content/117/11/6178/tab-figures-data. Source

## Abstract

Motile organisms actively detect environmental signals and migrate to a preferable environment. Especially, small animals convert subtle spatial difference in sensory input into orientation behavioral output for directly steering toward a destination, but the neural
mechanisms underlying steering behavior remain elusive. Here, we analyze a *C. elegans*
thermotactic behavior in which a small number of neurons are shown to mediate steering
toward a destination temperature. We construct a neuroanatomical model and use an evolutionary algorithm to find configurations of the model that reproduce empirical thermotactic
behavior. We find that, in all the evolved models, steering curvature are modulated by temporally persistent thermal signals sensed beyond the time scale of sinusoidal locomotion of
*C. elegans*. Persistent rise in temperature decreases steering curvature resulting in straight
movement of model worms, whereas fall in temperature increases curvature resulting in
crooked movement. This relation between temperature change and steering curvature
reproduces the empirical thermotactic migration up thermal gradients and steering bias
toward higher temperature. Further, spectrum decomposition of neural activities in model
worms show that thermal signals are transmitted from a sensory neuron to motor neurons
on the longer time scale than sinusoidal locomotion of *C. elegans*. Our results suggest that
employments of temporally persistent sensory signals enable small animals to steer toward
a destination in natural environment with variable, noisy, and subtle cues.

## Author summary

A free-living nematode *Caenorhabditis elegans* memorizes an environmental temperature
and steers toward the remembered temperature on a thermal gradient. How does the *C.
elegans* nervous system, consisting of 302 neurons, achieve the thermotactic steering
behavior? Here, we address this question through neuroanatomical modeling and simulation analyses. We find that persistent thermal input modulates steering curvature of

code used in the study is available at https://github.com/ikedamuneki/SteeringGA.

**Funding:** MI was supported by Grant-in-Aid for Scientific Research 16J05770 from the Ministry of Education, Culture, Sports, Science and Technology of Japan. This work was supported by NSF grant IIS-1845322 to EJI. The funders had no role in study design, data collection and analysis, decision to publish, or preparation of the manuscript.

**Competing interests:** The authors have declared that no competing interests exist.

model worms; worms run straight when they move up to a destination temperature, whereas run crookedly when move away from the destination. As a result, worms steer toward the destination temperature as observed in experiments. Our analysis also shows that persistent thermal signals are transmitted from a thermosensory neuron to dorsal and ventral neck motor neurons, regulating the balance of dorsoventral muscle contractions of model worms and generating steering behavior. This study indicates that *C. elegans* can steer toward a destination temperature without processing acute thermal input that informs to which direction it should steer. Such indirect mechanism of steering behavior is potentially employed in other motile organisms.

## Introduction

Animals sense environmental signals and navigate to a preferable environment [1,2]. Even when the distribution of signal is not known in advance, animals move around and detect a spatial signal gradient, enabling adjustments of moving direction for navigation. Such an active sampling of environmental signals is an essential component of the spatial navigation strategy for small animals, since it is difficult to detect a difference of signal intensity through multiple sensory organs placed on their tiny body [3,4].

With only a 1-mm-long body, the nematode *Caenorhabditis elegans* can sense and navigate gradients of gustatory, olfactory, and thermal signals [5–7]. During navigation, *C. elegans* makes gradual adjustments of its moving direction to steer upward/downward in a gradient [5,8,9]. Recent studies suggest a neural mechanism of steering behavior; worms could adjust the amplitude of head swings by sampling the difference of signal intensity through their own dorsoventral sinusoidal motion [10,11]. Supporting this hypothesis, oscillatory activity of pre-motor interneurons and motor neurons, which synchronizes with dorsoventral head swings, are modified upon the application of a favorable olfactory signal [12–14]. Further, optogenetic manipulation of a series of neurons only when the animal swings their head to dorsal or ventral side generates steering behavior to one direction [13,15,16]. However, during behavioral assays of freely moving animals, the difference of signal intensity sensed through head swings and resulting neural responses should be much more subtle. Especially in thermotaxis assays (**Fig 1A–1C**), temperature difference along one head swing is less than 0.01˚C on a linear thermal gradient of 0.5˚C/cm [7], which mimics natural thermal gradients in the upper few centimeters of soil [17]. Therefore, a novel mechanism potentially underlies steering behavior in such shallower signal gradients, albeit there had been no alternative hypotheses.

Here, we show that thermal inputs sensed not through head swings but through forward movement of *C. elegans* can generate steering behavior, leading worms to preferred temperature. To examine how the steering behavior is regulated during thermotaxis, we construct a neuroanatomically-grounded model with a set of neurons shown to mediate thermotactic steering behavior [9]. Thermal input sensed by the model worm was converted to the activity of a thermosensory neuron AFD through an empirical response property [18]; inter- and motor neurons were mathematically modeled as passive isopotential nodes with simple first order nonlinear dynamics [19,20]; and curving rates of locomotion were assumed to be proportional to the difference in activities of dorsoventral neck motor neurons. The unknown electrophysiological parameters of the model, including the sign and strength of the connections, were optimized by running a large set of evolutionary searches [21] so as to reproduce the empirical thermotactic behavior [9]. We found that in all the ensemble models obtained through evolutionary searches, steering curvature of model worms were modulated by the

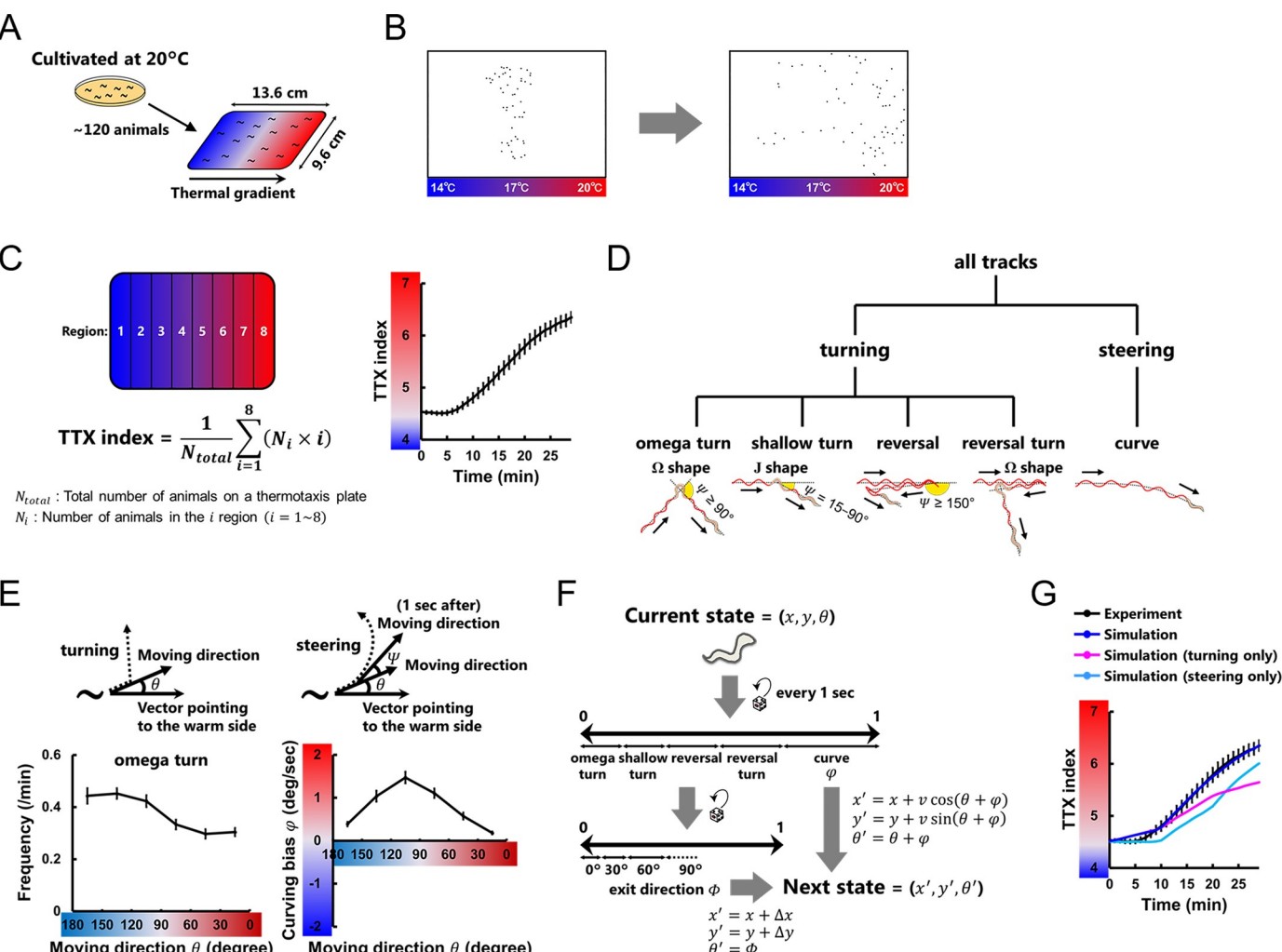

**Fig 1. Evaluation of *C. elegans* thermotactic behavior. (A)** Worms cultivated at 20°C migrate toward the cultivation temperature when placed on a plate with a thermal gradient [7]. **(B)** Representative thermotactic migration of approximately 120 worms that was recorded by a Multi-Worm Tracker [42]. **(C)** The time course of TTX indices (right panel) calculated using the described equation (left panel) and averaged within the assays (n = 12) [9]. **(D)** Classification and definition of *C. elegans* behavioral components used in this study. Turning and steering are classified as previously proposed [9]. **(E)** Frequency plot of the omega turns (left panel) and plot of the curving bias $\varphi$ (right panel) representing the averages as a function of the entry direction $\theta$ (upper panels) [9]. $\varphi$ is defined as $+\Psi$ if biased toward higher temperature and $-\Psi$ if biased toward lower temperature. **(F)** Schematic structure of the thermotactic simulation based on experimental data. Worm's state was defined by its position ($x$, $y$) and moving direction relative to the vector pointing to the warm side ($\theta$). We updated the states of the worm every second according to the experimentally observed data: the frequencies and the exit directions ($\Phi$) of turning, the curving biases ($\varphi$), and the locomotion speeds ($v$), all of which were applied as functions of $\theta$, temperature, and time (see Materials and Methods). The displacements during the individual turning ($\Delta x$, $\Delta y$) were also employed when updating the states of the worm. **(G)** The time course of TTX indices in experiments (black line) and simulations (colored lines) in which the data of turning or steering was replaced with the data of the corresponding component obtained in the experiments without thermal gradients [9]. In the individual simulation, we iterated assays 100 times, each with 100 animals, and the TTX indices were averaged within the assays. Error bars indicate SEM.

thermal input at the temporal scale of forward movements rather than the scale of head swings. Temperature rises through forward movements decreased steering curvature resulting in straight movement, whereas temperature falls increased curvature resulting in crooked movement. Our simulation analysis demonstrated that the observed relationship between temperature change and steering curvature can reproduce the empirical steering bias and thermotactic migration. Further, spectrum decomposition of neural activities in model worms showed that the dynamics of temperature signal was transmitted from a sensory neuron to motor neurons on a longer time scale than head swings. Our results suggest that the persistent signals sensed

through forward movement allow worms to adjust their moving direction toward a preferable environment, without knowing the specific steering direction during each head swing.

## Results

### Neuroanatomical models reproduce thermotactic steering behavior

*C. elegans* is known to navigate using a series of stereotyped movements: turning and steering [5,8,22] (**Fig 1D**). During thermotaxis assays (**Fig 1A–1C**), worms bias the frequency of turns according to their moving direction (**Fig 1E**), thereby migrating indirectly to a destination temperature. Worms also bias their curving rate (**Fig 1E**), thereby steering to a destination. A recent study conducted thermotactic simulation [9] in which states of worms were defined by their position in the assay plate ($x$, $y$) and their moving direction relative to the vector pointing to the warm side of the plate ($\theta$) (**Fig 1E and 1F**). At every step, the model worms were decided whether to perform turning or steering according to empirically observed probabilities. When turning, the next moving direction $\theta$ was provided based on empirical exit directions ($\Phi$); when steering, the next $\theta$ was provided based on empirical curving bias ($\varphi$). By conducting the simulations in which worms perform either biased turning or steering, the contribution of each of the strategies to the thermotactic migration were estimated [9] (**Fig 1G**).

To investigate how the nervous system implement the steering behavior during thermotaxis, we constructed a neuroanatomically-grounded model with a set of neurons shown to be involved in thermotactic steering behavior [9] (**Fig 2A**). In the model, thermal input sensed by a model worm was converted to the activity of a thermosensory neuron AFD through an empirical response property [18] (**Eq 1**), and inter- and motor neurons were mathematically modeled as passive isopotential nodes with simple first order nonlinear dynamics [19,20] (**Eq 3**). When model worms perform steering in the thermotactic simulation, the next moving direction $\theta$ was provided based on curving bias $\varphi$ calculated via the model circuit (**Fig 2A and 2B** and **Eqs 6–9**). The unknown parameters of the model were evolved using a genetic algorithm [21]; a large set of evolutionary searches were performed so that the thermotactic migration and steering behavior of model worms reproduced the empirical data (**Fig 2C**) (see Materials and Methods). Across 200 evolutionary searches, we obtained 8 independent parameter sets having a fitness score of at least 0.6 (**Figs 3A** and **S1B**). Although we did not find prominent common characteristics among the 8 sets of connection weights (**Figs 3B** and **S1C** and **S1 Table**), individual models reproduced the time course of TTX index and the curving bias observed in experiments (**Figs 3C** and **S1D**). Further, empirical impairments of curving bias in cell-ablated worms [9] were also reproduced in all the 8 models (**S1E Fig**). These results support that our models can serve as platforms to investigate how the neural circuit generates steering behavior during thermotaxis.

### Thermal input on the temporal scale of forward movement modulates steering curvature

Since there is no common characteristics among the connection weights of the models (**Figs 3B** and **S1C**), we first assessed whether there exist common profiles of steering behavior under simplified simulation settings. Temperature of assay plates was set as constant, and model worms were set not to perform turning. As shown in **Fig 4A**, the curvature $\Psi$ of model worm's trajectory was larger as the temperature of plates was higher. This positive relation between absolute temperature and steering curvature was observed in all the 8 parameter sets, though the magnitudes of curvature were diverse (**S5A Fig**). However, when the parameters were evolved in the simulation in which the response property of AFD was replaced with that of

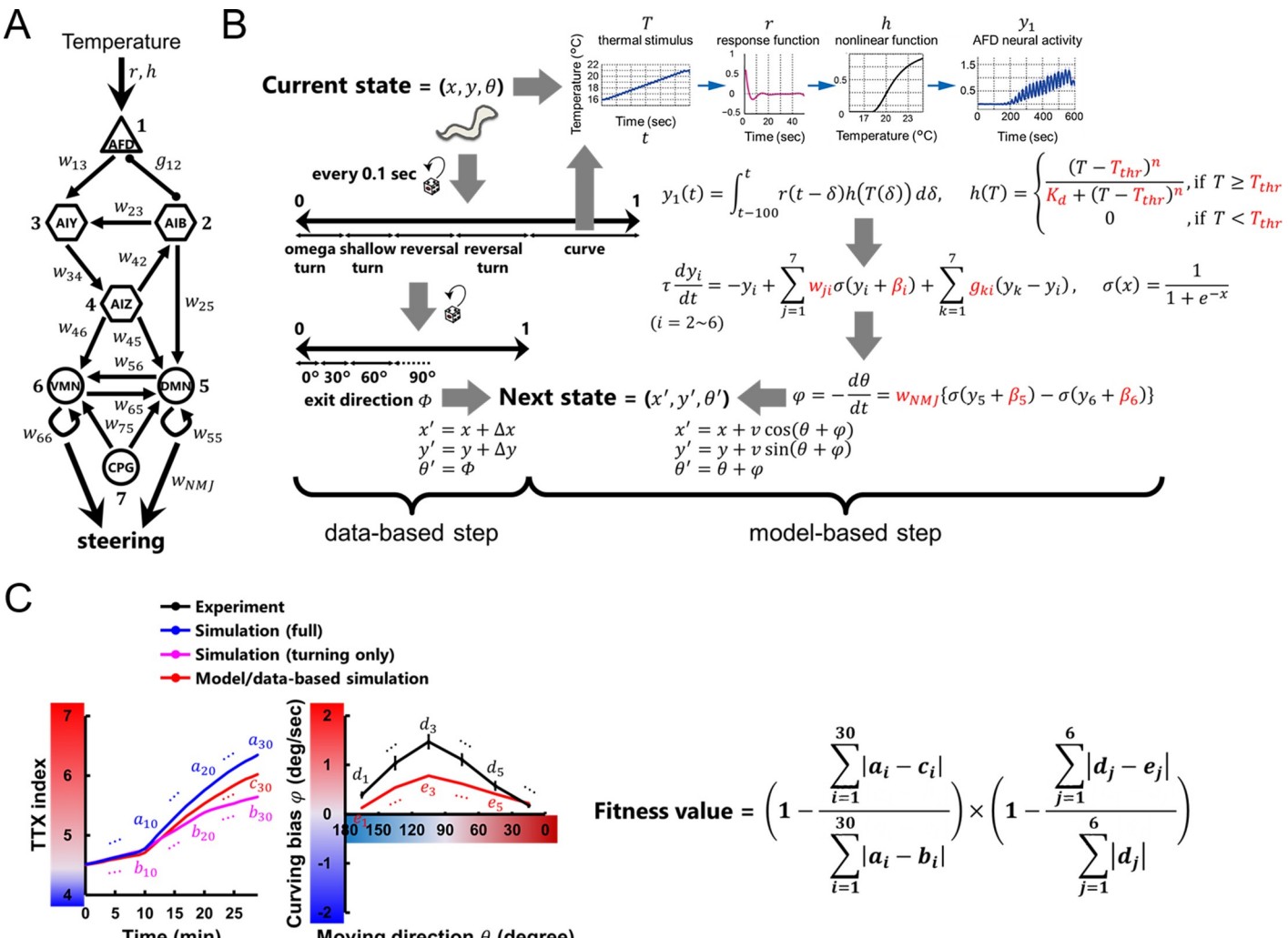

**Fig 2. Thermotactic simulation for building a neuroanatomical model.** **(A)** Neuroanatomical model that generates bias in steering behavior during positive thermotaxis [9], including thermosensory neurons (triangles), interneurons (hexagons), and head motor neurons (circles). Black thin arrows indicate chemical synapses, and black undirected lines with round endings gap junctions. An oscillatory component CPG is added to generate dorsoventral body undulation of *C. elegans*. **(B)** Schematic structure of the thermotactic simulation for searching parameters in the model with an evolution algorithm. Worm's state was defined by its position $(x, y)$ and moving direction $(\theta)$. We updated the states of the model worm every 0.1 second in two ways: according to the empirical data or via the neuroanatomical model for steering behavior. In the former case, frequencies, exit directions $(\Phi)$, and displacements $(\Delta x, \Delta y)$ during turning were applied as functions of $\theta$, temperature, and time. In the latter case, activity of AFD $(y_1)$ was estimated by empirically determined response property $r$ [18], activity of inter- and motor neurons $(y_{2-6})$ were calculated with simple first order nonlinear dynamics [40], and the magnitude of curving biases $(\varphi)$ was calculated proportionally to the difference in activities of dorsal and ventral neck motor neurons $(y_5$ and $y_6)$. Red parameters in the equations were optimized in evolutionary searches (see Materials and Methods). **(C)** Formula for the fitness value in evolutionary searches. TTX index and curving bias from the model/data-based simulations (red lines) were subtracted from the index in the data-based simulations (blue line in left panel) or the bias in the experiment (black line in meddle panel), respectively. The differences were summed up, normalized, and multiplied with each other to generate a total fitness value.

another sensory neuron [23] (**S2 Fig**), the curvature Ψ of the model worms was smaller as the temperature of plates was higher (**S5B Fig**). These results show that the relation between absolute temperature and steering curvature is not critical for reproducing the thermotactic behavior.

We next examined whether steering curvature are affected by derivatives of temperature. The model worms were exposed to temperature changes that mimic those sensed by the freely moving worms on a thermal gradient. One type of temperature change is at the temporal scale

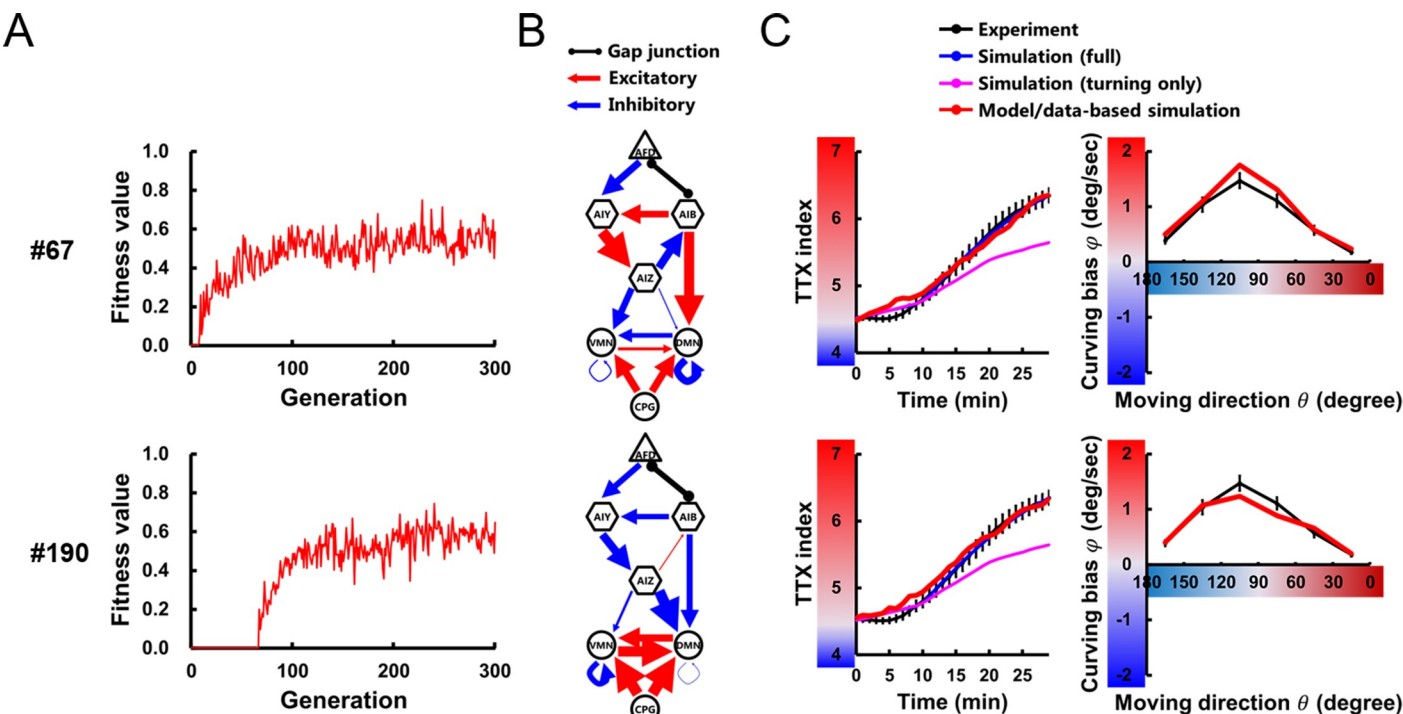

**Fig 3. Neuroanatomical models reproduce thermotactic behavior.** Through 200 evolutionary searches with 300 generations, 8 independent parameter sets that have fitness scores of at least 0.6 (**A**) were obtained (**S1 Fig**). Individual parameter sets were assigned numbers (#) from 1 to 200. For 2 of 8 models, the circuit diagram (**B**), the time course of TTX index, and the profile of curving bias (**C**) are plotted. In the circuit diagrams, thickness of each connection is represented proportionally to its connection weight (**S1 Table**).

of head swings of worms (**Fig 4B**). Due to their dorsoventral movement, worms sense sinusoidal changes in temperature of 1/4.2 Hz (**Eq 5**) with a maximum amplitude of 0.01˚C on a linear thermal gradient of 0.5˚C/cm [24]. Although the sensory neuron AFD in the models responded to the sinusoidal thermal input (**S6 Fig**), steering curvatures of the model worms were not different from those at the constant temperatures (**Fig 4A and 4B**). The other type of temperature change is at the temporal scale of forward movement of worms (**Fig 4C**). When moving straight up (or down) a thermal gradient of 0.5˚C/cm without turning nor steering, worms sense a monotonic temperature rise (or fall) of 0.01˚C/sec. We found that, within the range 18–20˚C, temperature rises decreased steering curvature of the model worms compared with those at a constant temperature, whereas temperature falls increased curvature within the range 16–20˚C. This relation between persistent changes in temperature and steering curvature Ψ was observed in all the 8 parameter sets (**S7A Fig**) and in the 4 parameter sets (**S7B Fig**) obtained with the different AFD response property (**S2 Fig**). Also, we additionally evolved moving velocity of model worms ($v$) and wave period of a pattern generator CPG ($t_{OSC}$) (**S3 Fig**; see Materials and Methods) and confirmed that the same relation between temperature changes and steering curvature Ψ was observed in different vales of $v$ and $t_{OSC}$ (**S7C Fig**). Further, when the parameters were evolved so that the model worms migrate down a thermal gradient [9] (**S4 Fig**), the opposite relation between temperature changes and Ψ was observed; the curvature of the model worms was larger (or smaller) as the temperature rises (or falls) (**S7D Fig**). These analyses suggest that thermal input during the worm's forward movement, not during the worm's head swings, modulates the curvature of locomotion, thereby steering to preferred temperature.

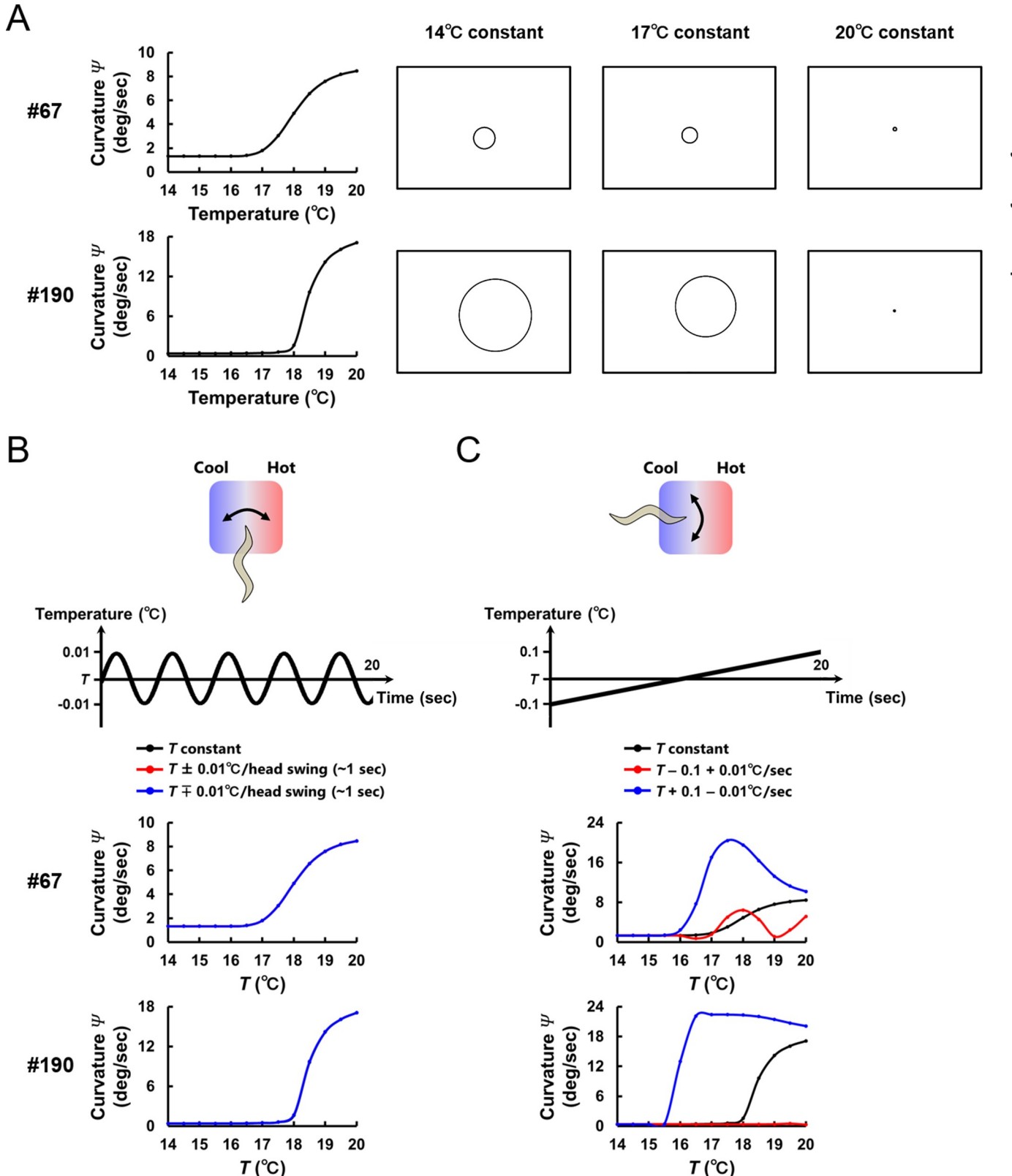

**Fig 4. Temperature change on the temporal scale of forward movement modulates curvature in locomotion. (A)** Trajectory of the model worms that were set not to perform turning on the constant temperatures ranging from 14 to 20˚C (right panels). The steering curvature Ψ of trajectories is calculated and plotted against

temperature (left panels). **(B)** Steering curvature Ψ under temperature change on the temporal scale of head swings (upper panels). Worms are assumed to be moving perpendicularly to a thermal gradient with their dorsal side heading toward warmer side (red lines) or colder side (blue lines). Ψ under these conditions were compared with those at the constant temperature (black lines). In the lower panels, red, blue, and black lines are overlapping. **(C)** Steering curvature Ψ under temperature change on the temporal scale of forward movement (upper panel). Worms are assumed to be moving straight up a thermal gradient (red lines) or down a thermal gradient (blue lines). Ψ under these conditions were compared with those at the constant temperature (black lines). The simulation results with representative parameter sets (#67 and #190) are shown.

## Thermotactic behavior is generated without directed biases in steering

The temperature changes employed in **Fig 4B and 4C** correspond to those sensed by worms moving perpendicularly or in parallel to a thermal gradient, in which the moving direction $\theta$ (**Fig 1E**) is equal to 90˚ or 0˚/180˚, respectively. We further examined steering curvature Ψ under other $\theta$ and represented the profiles of Ψ as functions of $\theta$ and temperature (**Fig 5A**). The positive relation between $\theta$ and Ψ was observed in all the 8 parameter sets (**S8A Fig**).

To assess whether the curvature profiles in **Fig 5A** can generate thermotactic migration and curving biases, we conducted another thermotactic simulation in which curving bias of model worms are decided not via the neuroanatomical model but based on the curvature profiles (**Fig 5B**). When model worms perform steering, the amplitude of curving bias ($|\varphi|$) is decided based on the profiles in **Fig 5A**, and the direction of steering (that is whether to steer higher or lower temperature direction) is randomly determined. Thus, the model worms lack the opportunity to bias steering consistently toward warmer or cooler directions. Nevertheless, we found that the simulations reproduced thermotactic migrations toward warmer direction and generated curving bias toward higher temperature (**Figs 5C** and **S8B**). This analysis demonstrates that the modulation of steering curvature upon thermal input on the temporal scale of forward movement (**Figs 4C** and **5A**) can generate thermotactic behavior.

## Higher activity of a thermosensory neuron induces straight movement of worms

The observation that steering curvature Ψ is dependent on temperature and moving direction $\theta$ (**Fig 5A**) implies the dependence of Ψ on activity of a thermosensory neuron AFD ($y_1$), since AFD encodes both absolute temperature and the differential of temperature [9,18,25]. To investigate this relation, we examined Ψ of model worms under another simplified simulation setting. AFD activity $y_1$ was fixed at constant value within the range in which model worms experience during the thermotactic simulations, and model worms were set not to perform turning. As shown in **Fig 6**, the curvature Ψ of model worm's trajectory was smaller as $y_1$ was higher. This negative relation was observed in all the 8 parameter sets, though some traces were non-monotonic (#39 and #124) (**S9A Fig**). Notably, in 4 of 8 parameter sets (#67, #88, #109 and #184), Ψ took the minimum values at specific $y_1$. The conversion from negative to positive relation was accompanied by the conversion of running direction from clockwise to counterclockwise, or vice versa (**S9B Fig**). Overall, AFD activity $y_1$ experienced by model worms during the thermotactic simulations covered the range in which Ψ showed dynamic decrease upon increase of $y_1$ and the values in which Ψ took the minimum (**S9A Fig**). These results suggest that thermal input sensed by AFD is efficiently transformed to regulate steering curvature, generating the curvature profiles (**Fig 5A**) and leading to thermotactic behavior (**Fig 5C**).

## Steering curvature is embedded in activity of motor neurons on a longer time scale than head swings

The next question to ask is how thermal input sensed by AFD is transmitted to inter- and motor neurons (**Fig 2A**) to generate steering behavior. Correlation analysis and information-

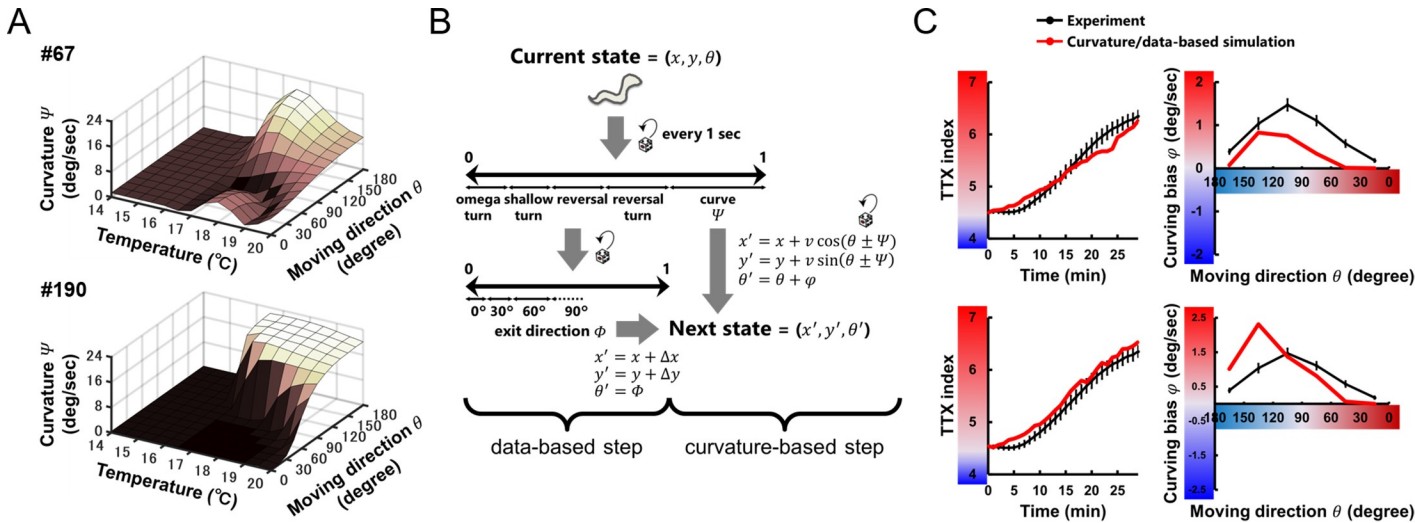

**Fig 5. Profiles of steering curvature reproduce thermotactic behavior. (A)** Steering curvature Ψ were calculated and plotted against temperature and moving direction θ. **(B)** Schematic structure of the thermotactic simulation based on the profile of steering curvature shown in **(A)**. We updated the states of the worm every 1 second according to the empirical data for turning [9] or via the profile of steering curvature Ψ, in which curving bias φ were calculated by multiplying random signs with Ψ. **(C)** The time course of TTX index (left panels) and the profile of curving bias (right panels) in experiments (black lines) and simulations (red lines) are plotted. The simulation results with representative parameter sets (#67 and #190) are shown.

theoretic analysis (see Materials and Methods) revealed that the dynamics of AFD activity $y_1$ is naively transmitted to amphid interneurons AIB, AIY, and AIZ (**Fig 7A and 7B**), in which transmission valences are consistent with connection valences (**Fig 3B**). By contrast, since oscillatory input from a central pattern generator (CPG) evokes oscillatory activity in dorsal/ventral motor neurons ($y_5$ and $y_6$) (**Fig 2A**) we did not observe a significant relation (linear or non-linear) between $y_1$ and $y_5/y_6$ (**Fig 7B**). However, $y_1$ and steering curvature Ψ showed similar relation with that observed in the simplified simulation (**Fig 6**). Since curving rates of

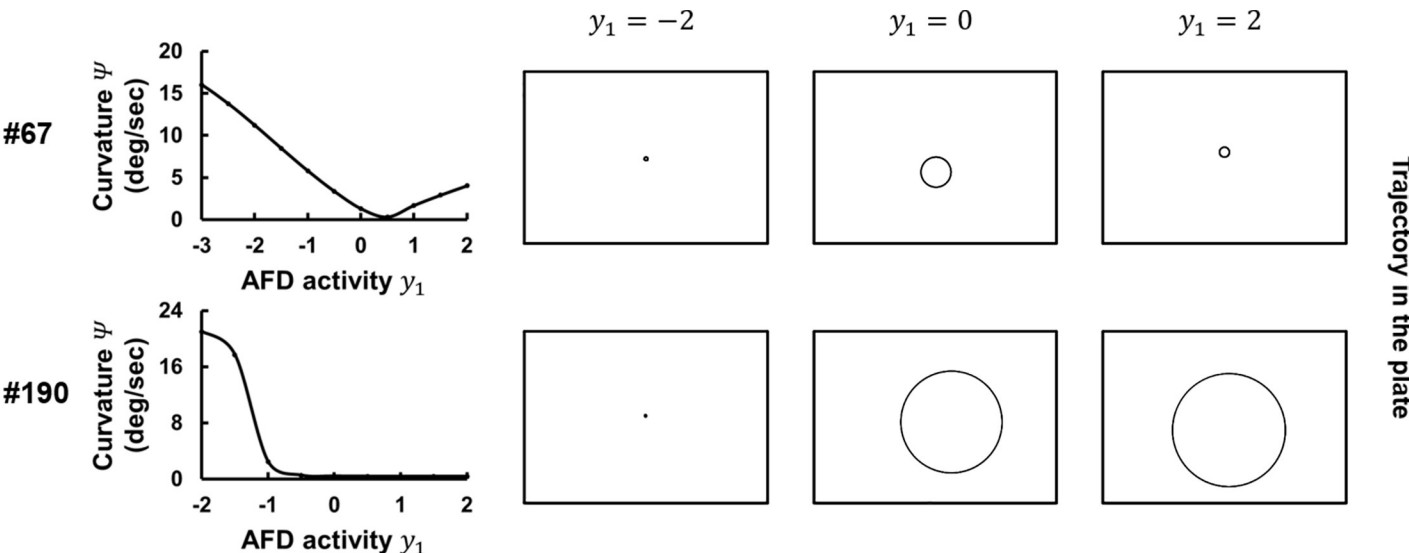

**Fig 6. Activity of a thermosensory neuron AFD designates steering curvature of locomotion.** Trajectory of the model worms on the fixed AFD activity $y_1$ (right panels). The steering curvature Ψ of trajectories is calculated and plotted against $y_1$ (left panels). The simulation results with representative parameter sets (#67 and #190) are shown.

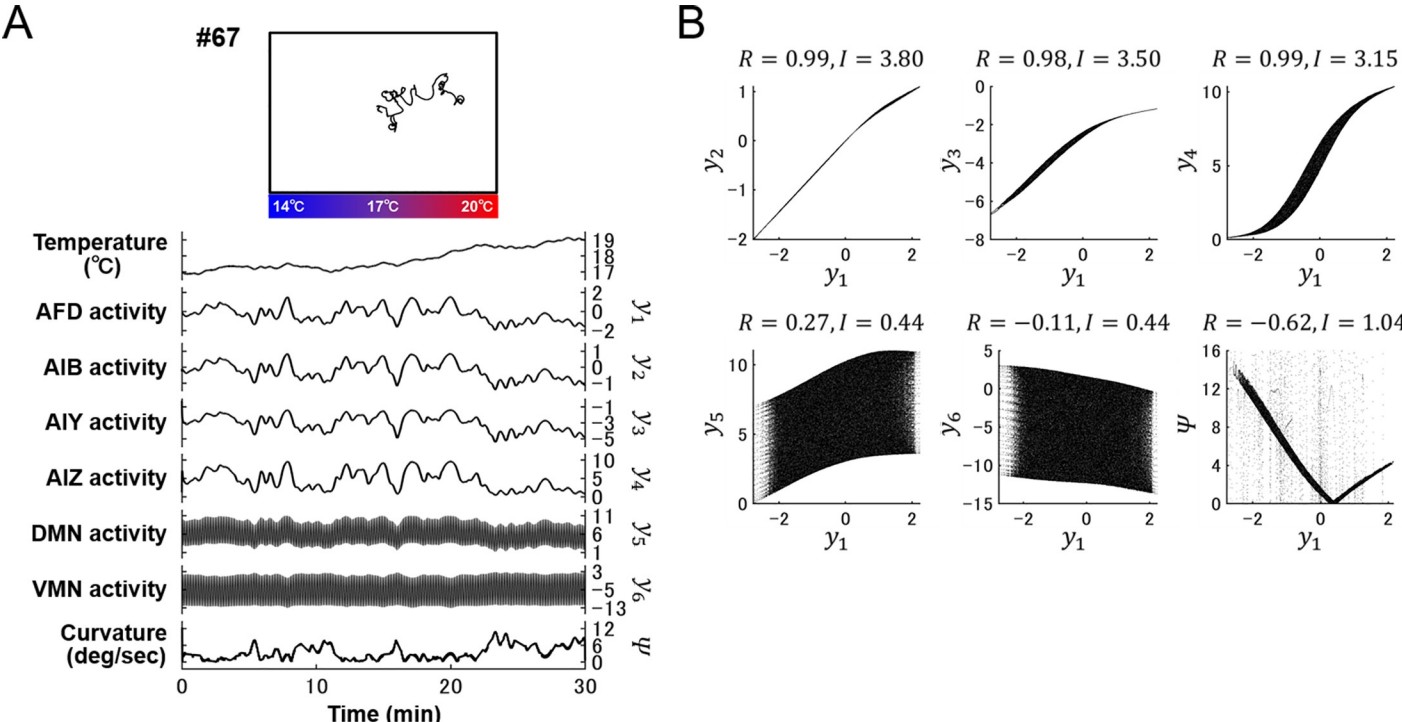

**Fig 7. Activity of a thermosensory neuron and interneurons shows strong correlations. (A)** Representative trajectory of a model worm (upper panel) generated through the neuroanatomical model with a representative parameter set (#67). The time series of temperature sensed by the worm, activity of individual neurons ($y_{1-6}$), and steering curvature of locomotion ($\Psi$) are represented in lower panels. **(B)** Scatter plots of AFD activity $y_1$ versus activity of other neurons $y_{2-6}$ or steering curvature $\Psi$ obtained from 100 model worms. Correlation coefficients ($R$) and mutual information ($I$) among the time series are measured (see Materials and Methods) and plotted in the panels.

model worms are calculated from $y_5$ and $y_6$ (**Eq 6**), these motor neurons somehow transmit upstream information into steering curvature.

To extract the components from $y_5$ and $y_6$ in which the dynamics of $y_1$ and $\Psi$ are embedded, we performed singular spectrum analysis (SSA). SSA decomposes time-series data into the left eigenvector (U) that corresponds to eigen-time-series (singular spectrum) and the right eigenvector (V) that represents the magnitude of each of the singular spectrum (**Fig 8A**; see Materials and Methods). SSA on the time courses of $y_5$ and $y_6$ (**Figs 7A and 8B**) revealed that the variables in V whose corresponding singular spectrum in U is constant within a 4-sec time window exhibited the strongest correlation ($R$) and mutual information ($I$) with AFD activity $y_1$ among other variables (**Fig 8C**). These variables of constant singular spectrum also exhibited the strongest $R$ and $I$ with steering curvature $\Psi$ (**Fig 8D**). This tendency was observed in all the 8 parameter sets (**S10C Fig**), in the 4 parameter sets obtained with the different AFD response property (**S10A and S10C Fig**), and in the 3 parameter sets with different vales of the worm's velocity and the wave period of CPG (**S10B and S10C Fig**). To further confirm the significance of non-oscillatory activity for executing steering behavior, we added a different time scales of noise component to the model circuits (**S11A Fig**) and assessed the effect on thermotactic simulations. As shown in **S11B Fig**, the applications of noise that oscillates with close frequency to that of CPG showed little or no impairments in the profiles of curving bias, whereas the applications of noise with slower oscillation impaired or diminished the curving bias toward higher temperature. These analyses indicate that the transmission of activity from sensory to motor neurons on the longer time scale than head swings of worms is crucial for thermotactic steering behavior.

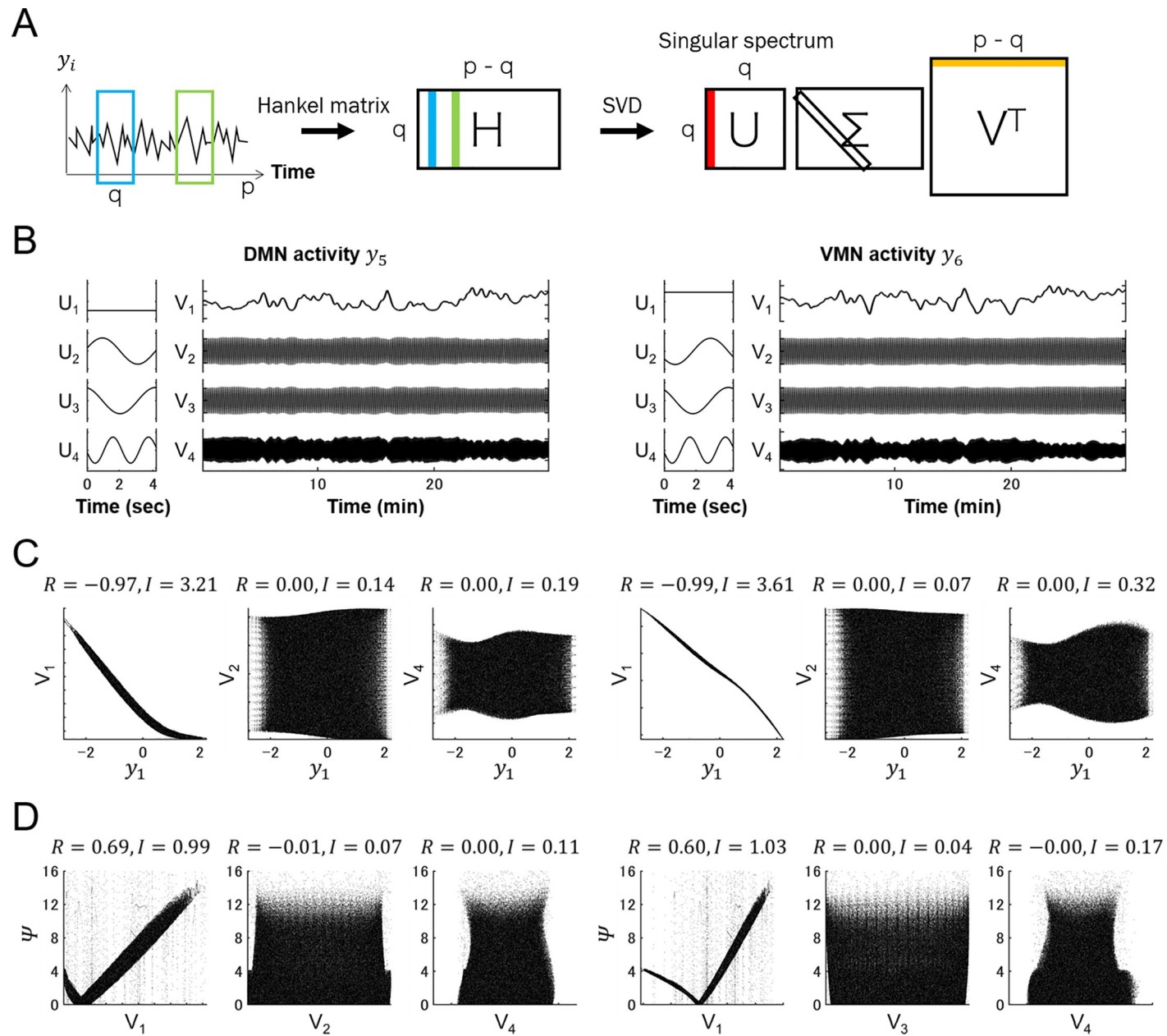

**Fig 8. Dynamics of constant singular spectrum of motor neuron activity show stronger correlations with a thermosensory neuron activity and steering curvature. (A)** Singular spectrum analysis (SSA) is an algorithm to decompose time-series data. A time series of neural activity data $y_i(t)$ ($i = 5,6; t = 1-p$) is stacked into a Hankel matrix (H) with a sliding time window of q. The singular value decomposition (SVD) of H yields a hierarchy of eigen-time-series that represent the singular spectrums (U) and their magnitude time series (V). **(B)** SSA was performed on activity of dorsal/ventral motor neurons ($y_5$ and $y_6$) of the model worm represented in **Fig 7A**. The first four singular spectrums $U_{1-4}$ and their magnitude $V_{1-4}$ are shown. **(C)** Scatter plots of AFD activity ($y_1$) versus each variable in $V_{1-4}$ obtained from 100 model worms. **(D)** Scatter plots of steering curvature $\Psi$ versus each variable in $V_{1-4}$ obtained from 100 model worms. In **(C)** and **(D)**, correlation coefficients ($R$) and mutual information ($I$) among the time series are measured and plotted in the panels. The simulation result with a representative parameter set (#67) is shown.

## Discussion

In this study, we constructed a *C. elegans* neuroanatomically-grounded model with a set of neurons that have been shown to mediate its thermotactic steering behavior (**Figs 1–3** and **7**). Simplified simulation analyses demonstrated that thermal input sensed not through head

swings but through forward movement of the model worms adjusts their steering curvature, thereby orienting to the preferred temperature (**Figs 4–6**). Singular spectrum analysis on generated neural activity data revealed that thermal input is transmitted from sensory to motor neurons over a longer time scale than dorsoventral head swings of the model worms (**Fig 8**).

Previous studies on *C. elegans* steering behavior have proposed that the adjustment of head swing amplitude during dorsoventral sinusoidal locomotion of *C. elegans* leads to direct navigation to a destination [10,11]. This type of orientation behavior is categorized as klinotaxis [1,26], and its implementation has been supported by neuroimaging [12–14], optogenetic experiments [13,15,16], and modeling studies in which neuroanatomical models were built to reproduce empirical chemotactic steering behavior [19,27]. By contrast, in this study, thermotactic steering behavior relied on the adjustment of steering curvature during forward locomotion (**Fig 4**), leading to indirect navigation to the destination temperature (**Fig 5**). This type of orientation behavior is categorized as klinokinesis [1,26], and klinokinetic implementation of steering is a potentially novel strategy for *C. elegans*, which has not been recognized previously. An advantage of klinokinetic steering is the reliability in environments with faint and noisy stimuli [28]. *C. elegans* lives in soil, where the temperature depends on the depth from soil surface, and the thermal gradient is assumed to be uniformly distributed with 0.5˚C/cm [17]. Thermal input sensed through *C. elegans* head swings in such environment is estimated to vary ±0.01˚C, with a low signal-to-noise ratio. Therefore, the longer time integration of thermal input might be necessary for executing steering behavior. By contrast, for larger animals such as *Drosophila* larva, thermal input sensed through its head casting can be sufficient for execute klinotactic thermotactic steering [29,30].

Further, it is possible that *C. elegans* chemotactic steering behavior is also implemented klinokinetically. Although previous studies of neuroimaging [12–14] and optogenetics [13,15,16] show that worms could adjust the amplitude of head swings and thus klinotactically steer, the possibility of klinokinetic steering has not been tested. The implementation of klinokinetic steering during thermotaxis and chemotaxis is potentially verified in assays in which thermal/chemical stimuli are applied temporally, and worm behavior is monitored simultaneously, as performed in **Fig 4**. By comparing steering curvature of worms under the application of sinusoidal or monotonic chemical/thermal stimuli, we could speculate which type of sensory input is more directly transformed to steering behavior. However, the application of sinusoidal stimuli with small amplitude and high frequency is still difficult to be achieved experimentally. Modeling studies are therefore crucial for investigating small animal behavior.

While our study proposed that monotonic thermal input during forward movement are responsible for steering up/down thermal gradients (i.e. positive/negative thermotaxis) (**Figs 4** and **S7**), we predict that oscillatory thermal input during sinusoidal locomotion can be employed for moving isothermally at around worm's cultivation temperature (i.e. isothermal tracking) [7,9]. A previous study showed that the amplitude of head swings during isothermal tracking were modulated not only by spatial but also temporal thermal gradients [24]. Also in chemical gradients, worms are reported to show horizontal locomotion along the edge of chemical distributions (i.e. surfing) [31]. Future implementation of isothermal tracking and surfing into neuroanatomical models will reveal the role of oscillatory sensory input during sinusoidal locomotion of *C. elegans*.

Neural activity data generated from the models underwent a decomposition analysis, called singular spectrum analysis (SSA) (**Fig 8**). Unlike other decomposition methods, such as short-time Fourier transform (STF) or continuous wavelet transform (CWT), SSA is a nonparametric decomposition of time series data [32,33] which does not assume temporal stationarity and spatial consistency within a multivariate system [34]. SSA decomposes time series into a sum of singular spectrums (U), which represent eigen-time-series, and time series vectors (V),

which represent the magnitude of each of the singular spectrum (**Fig 8A**). SSA was originally employed for denoising and extracting essential dynamics from measured data, especially geophysical time series [35–37]. A recent study utilized SSA for linear representation of nonlinear dynamics in chaotic systems [38]. Also, by investigating singular spectrums, SSA can be applied for change-point detection in time series [39]. In this study, we decomposed activity of the head motor neurons DMN/VMN and found that the magnitude of not oscillatory but constant spectrums exhibited the most evident relations with activity of the sensory neuron AFD and steering curvature $\Psi$ of the model worms (**Fig 8C and 8D**). Since DMN and VMN receive out of phase sinusoidal input from a pattern generator CPG (**Fig 2A**), their singular spectrums were mostly sinusoidal (**Figs 7A** and **8B**), which can be also extracted by STF or CWT. However, for example, the fourth spectrum of DMN in **S10B Fig** cannot be represented in these two methods. Most importantly, the dynamics that corresponds to constant spectrums cannot be naturally extracted by STF and CWT. Since the nervous system is temporally non-stationary and spatially inconsistent, SSA is potentially a powerful method for decomposing and understanding neural dynamics.

## Materials and methods

### Neuroanatomical modeling

Mathematical model of a neural circuit for regulating steering behavior (**Fig 2A**) was constructed based on previous studies [19,20], consisting of 7 neurons/components. Thermosensory neuron AFD (neuron ID $i = 1$) was modeled as a node with the response property $r$ that was identified previously [18] (**S2A Fig**). The response of AFD to temperature $T$ was obtained by linear convolution of input with $r$:

$$y_1(t) = \int_{t-100}^{t} r(t - \delta)h(T(\delta))d\delta \tag{1}$$

where $y$ represents the membrane potential (or neural response) at time $t$ relative to the resting potential (thus $y$ can assume positive and negative values). The Hill function

$$h(T) = \begin{cases} \dfrac{(T - T_{thr})^n}{K_d + (T - T_{thr})^n}, & \text{if } T \geq T_{thr} \\ 0, & \text{if } T < T_{thr} \end{cases} \tag{2}$$

represents the operating range of AFD, where $T_{thr}$ is the threshold temperature, $K_d$ is the dissociation constant, and $n$ is the Hill coefficient.

Interneurons and motor neurons ($i = 2–6$) were modeled as passive isopotential nodes with simple first order nonlinear dynamics [40]:

$$\tau \frac{dy_i}{dt} = -y_i + \sum_{j=1}^{7} w_{ji}\sigma(y_j + \beta_j) + \sum_{k=1}^{7} g_{ki}(y_k - y_i) \tag{3}$$

where $\tau$ is a time-constant, $\beta$ is a bias term that shifts the range of sensitivity of the output function, $w_{ji}$ is a strength of the chemical synapse from neuron $j$ to neuron $i$, and $g_{ki}$ is a conductance between neuron $k$ and neuron $i$. The first sum term is the input from the chemical synapses which were modeled as sigmoidal functions

$$\sigma(x) = \frac{1}{1 + e^{-x}} \tag{4}$$

of presynaptic voltage, and the second sum term is the input from the gap junctions which were modeled as non-rectifying conductances between two neurons.

The dorsal and ventral neck motor neurons receive out of phase input from an oscillatory component CPG ($i = 7$):

$$y_7(t) = \sin\left(\frac{2\pi t}{t_{OSC}}\right) \tag{5}$$

with $t_{OSC} = 4.2$ sec, which models dorsoventral body undulation of *C. elegans* on agar plates [41]. The curving rate $\psi$ deg/sec was calculated proportionally to the difference in activities of dorsal and ventral neck motor neurons ($i = 5, 6$):

$$\psi = w_{NMJ}\{\sigma(y_5 + \beta_5) - \sigma(y_6 + \beta_6)\} \tag{6}$$

where $w_{NMJ}$ is the strength of the connection from motor neurons to muscles. The circuit was simulated using the Euler method with a time step of 0.1.

## Thermotactic simulation

Thermotactic behavior was simulated with experimental data (**Fig 1E and 1F**) [9,42,43], combined with neuroanatomical models (**Fig 2A and 2B**) or profiles of steering curvature (**Fig 5A and 5B**). For each simulation, 100 worms were run sequentially. Worms were considered as dimensionless points in a 13.6 cm ($x$ axis) × 9.6 cm ($y$ axis) plate, with a linear thermal gradient from 14 to 20˚C or 20 to 26˚C along the $x$ axis. Worms started from the center of a plate, while $y$ coordinates and initial directions were randomized. For every 1 second or 0.1 second, individual model worms were decided whether to undergo an omega turn, a shallow turn, a reversal, a reversal turn, or a curve (**Fig 1D**). Event probabilities of each behavioral component were defined according to the experimental data of turning frequencies [9]. When model worms were decided to do any turns, the next positions ($x, y$) and moving direction $\theta$ were determined according to the experimental data of probability distributions of the exit direction $\Phi$ after the individual turns [9]:

$$\begin{cases} dx = \Delta x\, dt \\ dy = \Delta y\, dt \\ \quad \theta = \Phi \end{cases} \tag{7}$$

where ($\Delta x, \Delta y$) are the average displacements during the individual turns.

When model worms were decided to do a curve, the curving bias $\varphi$ was obtained in different ways depending on simulation types. In **Fig 1F**, $\varphi$ was obtained from the experimentally obtained profile (**Fig 1E**). In **Fig 2B**, the neuroanatomical model was employed to decide $\varphi$. Since moving direction $\theta$ is described relative to the vector pointing to the warm side of the plate (**Fig 1E**), curving rates $\psi$ (**Eq 6**) determine the change in $\theta$ differently depending on the dorsoventral direction $DV$ of the model worms:

$$d\theta = DV \times \psi\, dt \tag{8}$$

where $DV = +1$ (or $-1$) if the dorsal (or ventral) side of a model worm is toward the cold direction. Therefore, curving bias $\varphi$ was calculated as

$$\varphi = -\frac{d\theta}{dt} = -DV \times \psi \tag{9}$$

In **Fig 5B**, $\varphi$ was equal to $\pm\Psi$ whose sign was randomly determined in every step. In all the cases, the next positions $(x, y)$ were determined together with the moving velocity $v$ mm/sec:

$$\begin{cases} dx = v\cos(\theta + \varphi)dt \\ dy = v\sin(\theta + \varphi)dt \end{cases} \quad (10)$$

with $v = 0.2$ (or 0.3) mm/sec, which approximates the locomotion speed on a thermal gradient of 0.5˚C/cm whose temperature at the center is 17˚C (or 23˚C) [9].

Every experimental data was applied as a function of moving direction $\theta$. Besides, different data set were applied depending on whether worms were on the fraction 1–2, the fraction 3–6, or the fraction 7–8 of a thermotaxis plate (**Fig 1C**), and 0–10 min, 10–20 min, or 20–30 min of a simulation. If a worm reaches the plate border, it was set to do specular reflection.

## Evolution algorithm

Parameters of the neuroanatomical model were optimized by applying a genetic algorithm [21]. We optimized the following 23 parameters (ranges are shown in brackets) [44]: bias terms $\beta$ that shift sensitivity range of inter- and motor neurons (**Eq 3**) [−15, 15], weights of chemical synapses $w$ (**Eq 3**) [−15, 15], conductances of gap junctions $g$ (**Eq 3**) [0, 3], weight of neuromuscular junction $w_{NMJ}$ (**Eq 6**) [0, 90], and terms that determine response property of a thermosensory neuron (**Eq 2**): $T_{thr}$ [14, 26], $K_d$ [10, 000], and $n$ [1, 10]. In **S3 Fig**, we further optimized 2 more parameters: moving velocity $v$ of model worms (**Eq 10**) [0.01, 1] and wave period $t_{OSC}$ of a pattern generator CPG (**Eq 5**) [0.5, 50]. The optimization algorithm was run for populations of 96 independent parameter sets. Each time the algorithm was run, individuals were initialized by random selection. Populations were evolved for 300 generations. At the end of a run, the parameters of the best performing individual were stored for later analysis. The algorithm was run 200 times yielding 200 distinct model networks for **Figs 3** and **S1**, 100 times yielding 100 models for **S2 Fig**, 200 times yielding 200 models for **S3 Fig**, and 100 times yielding 100 models for **S4 Fig**.

Fitness of the model was evaluated based on thermotactic simulation (**Fig 2B**) in which the states of simulated animals were sequentially determined based on experimental data and neuroanatomical model (**Eqs 1–10**), yielding the time series of TTX index (**Fig 1C**) and the gross profile of curving bias $\varphi$ (**Fig 1E**). The TTX index was compared with those from data-based thermotactic simulation (**Fig 1F and 1G**), and the profile of curving bias was compared with that from experiment (**Fig 1E**). The fitness value was calculated as formalized in **Fig 2C**. For simplicity, negative values were set to zero.

## Behavioral analysis

Time series of the positions $(x, y)$ of simulated worms were analyzed following a previous study [9]. A mean filter within a moving 4.2 sec temporal window was applied to eliminate oscillatory components and estimate the trajectory of worm's centroid. For each frame, we defined the moving direction $\theta$ as the vectors from the current centroid to the following centroid (1 sec after), and calculated the steering curvature $\Psi$ by the angle between the previous moving direction (1 sec before) and the current moving direction (**Fig 1E**). Curving bias $\varphi$ was defined as $\Psi$ if the worm was steering toward higher temperature and $-\Psi$ if steering toward lower temperature. When yielding the profile of $\varphi$ as in **Fig 2C** or plotting $\Psi$ against other values as in **Fig 7B**, the frames of $\pm4.2$ sec within which the worm is performing turning behavior (**Fig 1D**) were eliminated from the analyses.

## Decomposition analysis

Time series of neural activity data $y_i(t)$ ($i$ = 5,6) underwent singular spectrum analysis (SSA) [32,33]. In SSA, we construct a Hankel matrix (H) from 1-d time series data:

$$H = \begin{bmatrix} y_i(t_1) & \cdots & y_i(t_{p-q+1}) \\ \vdots & \ddots & \vdots \\ y_i(t_q) & \cdots & y_i(t_p) \end{bmatrix} \tag{11}$$

where p is the length of time series, and q is the length of time windows (**Fig 8A**). The matrix H then underwent singular value decomposition (SVD):

$$H = U\Sigma V^T \tag{12}$$

where the left eigenvector U represents the feature vector of time-series, namely singular spectrum, and the right eigenvector V represents the magnitude of each of the singular spectrum.

## Information theoretic analysis

Relations among time series of neural activity data $y_i(t)$, magnitudes of singular spectrum $V_j(t)$, and steering curvature $\Psi(t)$ were assessed by measuring mutual information (Figs 7B and 8C and 8D) [45]. Mutual information is a measure of the dependence between two variables and quantifies the amount by which a measurement on one of the variables reduces our uncertainty about the other. Calculation of mutual information was performed as previously described [27]. The discrete probability distributions of time-series data were estimated over a fine grid of 50 bins and by a kernel density estimation technique known as average shifted histograms [46], with 12 shifts along each dimension. Mutual information can measure a non-linear relation between variables, while Pearson's correlation coefficient measures a linear relation.

## Quantification

Experimental data are expressed as mean ± SEM. Simulation data are expressed as mean.

## Supporting information

**S1 Fig. Thermotactic behavior is reproduced by a variety of neuroanatomical models. (A)** 32 independent parameter sets having a fitness score of at least 0.5, including 8 parameter sets having a fitness score of at least 0.6 **(B)** are plotted in parameter spaces as black dots and red dots, respectively. The parameter subspaces are the space defined by principal components of 6 bias terms that shift sensitivity range of inter- and motor neurons (leftmost panel), the space defined by principal components of 12 connection weights of chemical/electrical synapses (second left panel), the plane defined by a connection weight from a pattern generator to motor neurons and a connection weight of neuromuscular junction (second right panel), and the space defined by 3 terms that determine response property of a thermosensory neuron (rightmost panel). Individual parameter sets were assigned numbers (#) from 1 to 200. For the 8 good models, the circuit diagram **(C)**, the time course of TTX index, and the profile of curving bias **(D)** are plotted. In the circuit diagrams, thickness of each connection is represented proportionally to its connection weight. All the 8 models reproduced empirical impairments of curving bias upon ablating individual interneurons AIB, AIY, and AIZ **(E)**.
(TIF)

**S2 Fig. Thermotactic behavior is reproduced with a different response property of a sensory neuron.** We performed 100 evolutionary searches in which the response property of AFD [18] was replaced with that of another sensory neuron AWC [23] **(A)**, and 4 independent parameter sets having a fitness score of at least 0.5 were obtained **(B)**. Individual parameter sets were assigned numbers (#) from 201 to 300. For the 4 good models, the circuit diagram **(C)**, the time course of TTX index, and the profile of curving bias **(D)** are plotted. In the circuit diagrams, thickness of each connection is represented proportionally to its connection weight. (TIF)

**S3 Fig. Specific magnitude of worm's velocity is preferred for reproducing thermotactic behavior.** We performed 200 evolutionary searches in which moving velocity of model worms ($v$) and wave period of a pattern generator CPG ($t_{OSC}$) were evolved, and 68 independent parameter sets having a fitness score of at least 0.5 were obtained **(A and B)**. Individual parameter sets were assigned numbers (#) from 301 to 500. For the 3 representative models (red dots in **(A)**), the circuit diagram **(C)**, the time course of TTX index, and the profile of curving bias **(D)** are plotted. In the circuit diagrams, thickness of each connection is represented proportionally to its connection weight. (TIF)

**S4 Fig. Negative thermotactic behavior is also reproduced by a variety of neuroanatomical models.** We performed 100 evolutionary searches in which the parameters were evolved to reproduce negative thermotactic behavior **(A)**, and 8 independent parameter sets having a fitness scores of at least 0.6 were obtained **(B)**. Individual parameter sets were assigned numbers (#) from 501 to 600. For the 8 good models, the circuit diagram **(C)**, the time course of TTX index, and the profile of curving bias **(D)** are plotted. In the circuit diagrams, thickness of each connection is represented proportionally to its connection weight. (TIF)

**S5 Fig. Relation between temperature and steering curvature of model worms.** Steering curvature $\Psi$ of the model worms were measured under the simplified simulation in which temperature of assay plates was set as constant (ranging from 14 to 20˚C or from 20 to 26˚C), and model worms were set not to perform turning. The parameter sets evolved through different evolutionary searches were employed for the simulation, and steering curvature $\Psi$ of the individual model worms are plotted against temperature: **(A)** for **S1 Fig**, **(B)** for **S2 Fig**, **(C)** for **S3 Fig**, and **(D)** for **S4 Fig**. (TIF)

**S6 Fig. AFD responds to both temperature changes on the temporal scale of head swings and forward movement. (A)** Activity of AFD (red line) under temperature changes on the temporal scale of head swings (black line). Worms are assumed to be moving perpendicularly to a thermal gradient with their dorsal side heading toward warmer side. **(B)** Activity of AFD (red line) under temperature changes on the temporal scale of forward movement (black line). Worms are assumed to be moving straight up a thermal gradient. The simulation results with representative parameter sets (#67 and #190) are shown. (TIF)

**S7 Fig. Temperature change on the temporal scale of forward movement modulates steering curvature of locomotion.** Steering curvature $\Psi$ of the model worms were measured under temperature change on the temporal scale of head swings (left panels) and of forward movement (right panels). In the left panels, worms are assumed to be moving perpendicularly to a thermal gradient with their dorsal side heading toward warmer side (red lines) or colder side

(blue lines). In the right panels, worms are assumed to be moving straight up a thermal gradient (red lines) or down a thermal gradient (blue lines). Ψ under these conditions were compared with those at the constant temperature (black lines). The parameter sets evolved through different evolutionary searches were employed for the simulation: **(A)** for **S1 Fig**, **(B)** for **S2 Fig**, **(C)** for **S3 Fig**, and **(D)** for **S4 Fig**.
(TIF)

**S8 Fig. Profiles of steering curvature reproduce thermotactic behavior. (A)** Steering curvature Ψ were calculated and plotted against temperature and moving direction $\theta$. **(B)** Plots of the time course of TTX index (left panels) and the profile of curving bias $\varphi$ (right panels) in experiments (black lines) and simulations (red lines) in which the states of the worm are updated according to the empirical data for turning and to the profile in **(A)** for steering.
(TIF)

**S9 Fig. Activity of a thermosensory neuron AFD modulates curving rates and circulating direction of locomotion. (A)** Steering curvature Ψ of the model worms were measured under the simplified simulation in which AFD activity $y_1$ of the model circuits was fixed at the constant values, and model worms were set not to perform turning. **(B)** Trajectories of the model worm with representative parameter set (#88) on the fixed AFD activities.
(TIF)

**S10 Fig. Dynamics of constant singular spectrum of motor neuron activity show stronger correlation with a thermosensory neuron activity and steering curvature. (A and B)** Upper panels are time series of temperature sensed by the model worms evolved in **S2(A) Fig** and **S3 (B) Fig**, activity of individual neurons $y_{1-6}$, steering curvature Ψ of locomotion, first four singular spectrums $U_{1-4}$ of dorsal motor neuron activity $y_5$ decomposed by singular spectrum analysis, and their magnitude $V_{1-4}$. Scatter plots of each variable in $V_{1-4}$ versus AFD activity $y_1$ or steering curvature Ψ are shown in lower panels, where correlation coefficients $R$ and mutual information $I$ among the time series are measured and plotted. **(C)** Plots of $R$ and $I$ measured among variable in $V_{1-4}$ versus AFD activity $y_1$ and steering curvature Ψ in all the 15 good/representative models evolved in **S1–S3** Figs.
(TIF)

**S11 Fig. Application of oscillatory noise component to the model circuits impairs thermotactic steering behavior. (A)** A different time scales of noise component are added in the representative model circuits. The noise component is set to transmit oscillatory input to either AIY or DMN, and the wave period of the oscillation was changed from that of DV head bending ($t_{OSC}$) to 10 times longer value. **(B)** Profiles of curving bias $\varphi$ of the model worms under the applications of the oscillatory noise component to AIY (upper panels) or DMN (lower panels).
(TIF)

**S1 Table. Parameters and values optimized in evolutionary searches.** Values of 23 (or 25) parameters in the 23 good/representative models evolved in **S1–S4** **Figs** are listed.
(TIF)

# Acknowledgments

We would like to thank Ikue Mori for supervising experimental data acquisition; Amane Kano and Yuki Tsukada for providing code to estimate neural responses; Erick Olivares and Randall D. Beer for valuable comments on modeling analyses; and Jun Kitazono and Masafumi Oizumi for helpful discussions.

## Author Contributions

**Conceptualization:** Muneki Ikeda, Eduardo J. Izquierdo.

**Data curation:** Muneki Ikeda.

**Formal analysis:** Muneki Ikeda, Hirotaka Matsumoto.

**Funding acquisition:** Muneki Ikeda, Eduardo J. Izquierdo.

**Investigation:** Muneki Ikeda, Hirotaka Matsumoto, Eduardo J. Izquierdo.

**Methodology:** Muneki Ikeda, Hirotaka Matsumoto, Eduardo J. Izquierdo.

**Project administration:** Muneki Ikeda, Eduardo J. Izquierdo.

**Resources:** Muneki Ikeda, Eduardo J. Izquierdo.

**Software:** Hirotaka Matsumoto, Eduardo J. Izquierdo.

**Supervision:** Eduardo J. Izquierdo.

**Validation:** Muneki Ikeda, Hirotaka Matsumoto, Eduardo J. Izquierdo.

**Visualization:** Muneki Ikeda, Hirotaka Matsumoto.

**Writing – original draft:** Muneki Ikeda.

**Writing – review & editing:** Muneki Ikeda, Hirotaka Matsumoto, Eduardo J. Izquierdo.

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
