## [Decision Letter · Decision Letter 0]

11 Aug 2020

Dear Dr. Ikeda,

Thank you very much for submitting your manuscript "Persistent thermal input controls steering behavior in Caenorhabditis elegans" for consideration at PLOS Computational Biology.

As with all papers reviewed by the journal, your manuscript was reviewed by members of the editorial board and by several independent reviewers. In light of the reviews (below this email), we would like to invite the resubmission of a significantly-revised version that takes into account the reviewers' comments.

We cannot make any decision about publication until we have seen the revised manuscript and your response to the reviewers' comments. Your revised manuscript is also likely to be sent to reviewers for further evaluation.

Sincerely,

Tosif Ahamed

Guest Editor

PLOS Computational Biology

Lyle Graham

Deputy Editor

PLOS Computational Biology

Reviewer's Responses to Questions

**Comments to the Authors:**

Reviewer #1: The authors of this manuscript used neuroanatomical information available for the temperature sensing and response circuit in the nematode C. elegans to produce a (empirically-informed) model for thermotaxis in C. elegans. They use evolutionary searches to find patterns of connection strength and sign between the AFD, AIB, and AIY neurons (among others) capable of modeling observed steering rates. The authors found that steering rates were modulated by inputs at scales consistent with forward locomotion (rather than the faster oscillatory head swings).

I thought that the manuscript was well written and that the results presented supported the conclusions drawn. The information contained in this effort should prove useful to scientists interested in sensory integration in general, and thermotaxis in particular. I have a few comments that are mostly directed to the framing of the work, rather than its design and performance.

1) The work undertaken is restricted to animals migrating up a thermal gradient towards a cultivation temperature (except for the isothermal experiments). However, work on C. elegans on animals escaping noxious temperature stimuli, or down thermal gradients towards cultivation temperatures. Did the authors attempt these scenarios and how do they results reconcile with those behavioral scenarios?

2) The idea that longer time integration is preferred seems plausible for an organism that locomotes through sinusoidal waves such as C. elegans. Is there any thoughts on the interaction between sideways and forwards locomotion in the production of the model? For example, if younger (smaller) worms (with different sinusoidal excursion and forward velocity) were used, would the plots on Fig 4 B and C now look different? In other words, would the model scale across animal sizes, or would you predict it to be best suited across specific forward velocity and head bend frequency ranges?

3) The authors suggest that in addition to using klinotaxis for chemical orientation, worms use klinokinetics for thermal orientation. Could they please discuss the advantages and limitations for each sensory modality to the task of navigation, in a way that explained how each would necessitate its own behavioral mechanism?

Smaller points:

1) The meaning of the boxes on the right side of Figure 6 or the right panel A of Figure 4 are only clear after reading the caption to Figure S5B. I would make the caption on these figures clearer.

2) In Figure 7B why is the correlation between AFD and DMN and VMN not symmetrical? Is there a dorsal or ventral bias?

3) I found Figure 8 is confusing/challenging (perhaps due to my unfamiliarity with the algebra). I think that from the perspective of a wider audience this figure could be further digested for the reader (perhaps in a supplementary figure).

4) Line 43: C. elegans does not possess a 302-neuron brain. Rather, its entire nervous systems is comprised of 302 neurons.

5) The algorithms used and the raw data generated (both from the behavioral experiments as well as the modeling data) was not deposited in an accessible repository (although the supplement does provide graphical summaries). I am not sure if the journal does or not require these to be submitted.

Reviewer #2: Ikeda et al., present an interesting study that nicely uses the advantages of mathematical modeling to disentangle whether the sinusoidal variation in sensory input imposed by the wiggling motion of the worm contributes to behavioral mechanisms for thermotaxis. They do this by developing 9 model worms using genetic algorithms and then testing how the worm responds to controlled sensory stimuli when aspects of motor response were turned off. They found that the models did not respond differently when sensory input was made sinusoidal. They also investigated how the activity of the model sensory neuron contributed to curving rates using singular spectrum analysis. This supported the idea that information transferred on the time scale longer than head bends is most important.

Major concerns:

The study tests a major hypothesis of whether the weathervane behavioral mechanism contributes to thermotaxis behavior in C. elegans, or just a version of the pirouette mechanism. Evidence for the weathervane mechanism has been found for

1. Ln 145. “on the scale of forward movement” and Ln 164 “scale of forward movement of worms” and other places – what exactly is this scale? It is not clear in this section what time scales and change-in-temperate rates you are comparing. Please explicitly explain these values side by side, and put in context with the durations of DV head bends and forward runs. Forward movement during runs has huge variation. Do you mean to conclude that the temporal scale important for causing steering maneuvers is X times longer than the duration of DV head bends?

2. If oscillatory input during DV head bends does not modulate curving rates, does it modulate pirouettes?

3. SSA correlation analysis provided one type of evidence that thermal info propagated from sensory neuron to motor neurons on a time scale longer than the DV head bending for the 9 models. It would be great to complement this correlational analysis with experimental analysis that modifies the networks on short and long time scales to independently test which time scale is most important and how. For instance, does filtering or adding noise on different time scales in the circuit interfere with performance? Does this support the idea that input on time scale of DV head bends doesn’t matter?

Minor concerns:

A.) There are many places where the text would benefit from editing, and defining key terms and ideas as they are first introduced. These include:

Abstract

Ln 21. Small animals, not small-size

Ln 21. Subtle *difference* in sensory input – what kind of “difference”? Temporal, spatial, or both? And/or modality? You don’t explain.

Ln 28. Persistent thermal signals – it is critical to explain this better. Persistent in modality? Same modality but integrated across a period of time? How persistent? How much time? And persistent across time relative to what? DV head bends? And/or beyond? Context is important here.

Ln 29. Lessens steering rates – unless “steering” is defined otherwise later in the paper, readers will know that one can “steer” straight, so the wording is confusing. How about “turn rates”?

Ln 29. *Persistant temperature increment* lessens.. this is awkward phrasing. Do you mean persistent changes in temperature? If so, then changes from what? Change and increment are all relative.

Ln 171. …thermal input DURING the worm’s forward movement…

B.) Although no similarity was found for the values underlying the 9 models, it would be nice to hear how clustered the models were in search space.

Reviewer #3: See attachment

**Have all data underlying the figures and results presented in the manuscript been provided?**

Reviewer #1: **No: **no raw behavioral data, nor modeling output data was provided. No algorithm provided

Reviewer #2: Yes

Reviewer #3: **No: **Raw data was not made available in the current version of the manuscript

PLOS authors have the option to publish the peer review history of their article (what does this mean?). If published, this will include your full peer review and any attached files.

Reviewer #1: No

Reviewer #2: No

Reviewer #3: No
---

## [Decision Letter · Decision Letter 1]

17 Nov 2020

Dear Dr. Ikeda,

We are pleased to inform you that your manuscript 'Persistent thermal input controls steering behavior in Caenorhabditis elegans' has been provisionally accepted for publication in PLOS Computational Biology.

Best regards,

Tosif Ahamed

Guest Editor

PLOS Computational Biology

Lyle Graham

Deputy Editor

PLOS Computational Biology

Reviewer's Responses to Questions

**Comments to the Authors:**

Reviewer #1: In this revised version of their manuscript Ikeda et al made significant efforts to address the comments provided by this (and other) reviewers. I feel the manuscript reads better and the figures are clearer than in the first iteration. I am satisfied that the authors have taken this opportunity to improve their work. I note that the codes used are now available as per journal requirements. This work contributes to our understanding of temperature integration by C. elegans, but I suspect it will also prove useful to understanding how other organisms deal with inputs that challenge the detection limits of other sensory apparatuses.

Reviewer #2: The authors addressed all of my concerns.

Reviewer #3: I find the revised manuscript significantly improved from the initial submission, both in terms of scientific rigor and clarity of presentation. The revised manuscript now better distinguishes the contribution of steering vs turning in the thermotaxis performance of the evolved models, which was my main concern previously. I also appreciate the authors’ efforts in performing several computational controls to provide better intuition on what features of the evolved networks contributed to efficient thermotaxis. Overall, I think the revised manuscript presents a novel computational analysis on a navigation behavior commonly observed across organisms.

**Have all data underlying the figures and results presented in the manuscript been provided?**

Reviewer #1: Yes

Reviewer #2: Yes

Reviewer #3: Yes

PLOS authors have the option to publish the peer review history of their article (what does this mean?). If published, this will include your full peer review and any attached files.

Reviewer #1: No

Reviewer #2: No

Reviewer #3: No

---

## [Editor Report · Acceptance letter]

6 Jan 2021

PCOMPBIOL-D-20-00676R1 

Persistent thermal input controls steering behavior in Caenorhabditis elegans

Dear Dr Ikeda,

I am pleased to inform you that your manuscript has been formally accepted for publication in PLOS Computational Biology. Your manuscript is now with our production department and you will be notified of the publication date in due course.

With kind regards,

Jutka Oroszlan
